# Swift and extensive Omicron outbreak in China after sudden exit from 'zero-COVID' policy

Emma E. Goldberg[1], Qianying Lin [1], Ethan O. Romero-Severson [1] & Ruian Ke [1] ✉

In late 2022, China transitioned from a strict 'zero-COVID' policy to rapidly abandoning nearly all interventions and data reporting. This raised great concern about the presumably-rapid but unreported spread of the SARS-CoV-2 Omicron variant in a very large population of very low pre-existing immunity. By modeling a combination of case count and survey data, we show that Omicron spread extremely rapidly, at a rate of 0.42/day (95% credibility interval: [0.35, 0.51]/day), translating to an epidemic doubling time of 1.6 days ([1.6, 2.0] days) after the full exit from zero-COVID on Dec. 7, 2022. Consequently, we estimate that the vast majority of the population (97% [95%, 99%], sensitivity analysis lower limit of 90%) was infected during December, with the nation-wide epidemic peaking on Dec. 23. Overall, our results highlight the extremely high transmissibility of the variant and the importance of proper design of intervention exit strategies to avoid large infection waves.

From the beginning of the SARS-CoV-2 pandemic until fall 2022, China maintained a strict 'zero-COVID' set of policies that implemented strong non-pharmaceutical interventions such as city-wide lock-downs, travel and movement restrictions, contact tracing of direct and secondary contacts, and compulsory quarantine of infected individuals and their contacts in centralized facilities. In addition, to track the spread of the virus, the population was tested on a regular bases (every 2–4 days) using polymerase chain reaction (PCR) tests. As a result, the number of SARS-CoV-2 infected individuals was kept at very low levels. Beginning on Nov. 11, 2022, control measures were rapidly relaxed. First during Nov. 11 and Dec. 7, the 20 Measures policy was implemented where some of the strict intervention efforts were relaxed[1]. Under this policy, city-wide lockdowns were replaced with targeted lock-downs to areas and buildings where infections were detected, travel restrictions were partially lifted, and mass testing and reporting were no longer compulsory. Contact tracing was limited to direct contacts of infected persons only, and the quarantine period was shortened. Starting from Dec. 7, 10 Measures replaced 20 Measures and nearly all intervention efforts of the zero-COVID policy were removed, fully exiting from the zero-COVID policy. During this period, lock-down was prohibited, contact tracing was stopped, and

quarantine at centralized location was replaced by home isolation, etc.[2]. Testing and reporting became voluntary. This abrupt exit from zero-COVID raised public health concerns about the unchecked spread of SARS-CoV-2[3], especially given the high transmissibility of the Omicron variants BA.5 and BF.7 present in China[4,5]. Indeed, a few weeks after the full exit from zero-COVID, it was reported that high numbers of patients with respiratory illness were overwhelming hospitals in China[6]. Furthermore, on three flights from China to Italy in late Dec. 2022, ~40%–50% of passengers tested positive for COVID-19[7,8]. These observations strongly suggest that SARS-CoV-2 was already widespread in China by the end of Dec. 2022, in contrast to the official data that showed daily cases during December to be low and waning (Fig. 1a).

Accurately quantifying the dynamics of the Omicron outbreak in China is valuable for several reasons. First, it is essential to understand the magnitude of the public health crisis induced by large-scale SARS-CoV-2 spread in a population of size >1.4 billion. Second, it will reveal the efficacy of the previously strict non-pharmaceutical intervention efforts during late Oct. and early Nov. in China. This in turn can help to design effective combinations of pharmaceutical and non-pharmaceutical intervention policies to better 'flatten the curve' of

[1]Theoretical Biology and Biophysics (T-6), Los Alamos National Laboratory, Los Alamos, NM 87545, USA. ✉e-mail: rke@lanl.gov

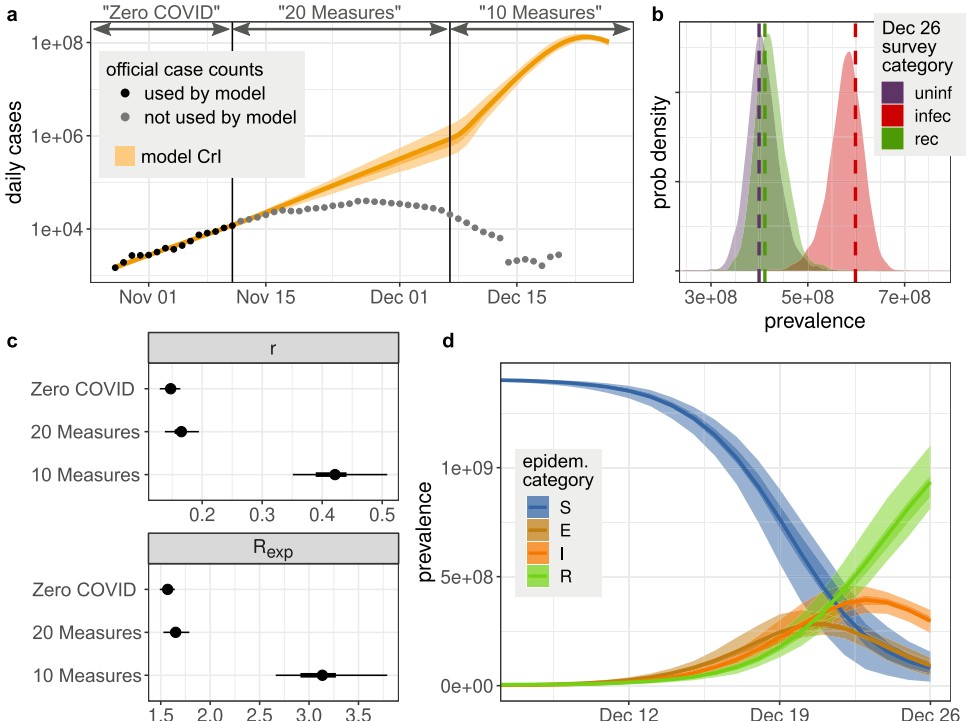

**Fig. 1 | Model-predicted epidemic dynamics of the Omicron variant in China.**
**a** The model-inferred daily cases (orange bands) during the three policy periods (zero-COVID, 20 Measures, and 10 Measures) near the end of 2022, and official case counts (points). Shades of orange show the median, 50% credibility interval (CrI), and 95% CrI. **b** Survey results, interpreted as population size, compared against model-predicted values for those survey categories on Dec. 26. Dashed lines are the data, and shaded curves are the posterior distribution of expected number of individuals. **c** Estimates of the intrinsic rate of increase, $r$, and reproductive number, $R_{exp}$ during the exponential growth of each time period. From the model fit, points are the median, thick lines are the 50% CrI, and thin lines are the 95% CrI. **d** Model-inferred epidemic dynamics during 10 Measures. The median, 50% CrI, and 95% CrI are shown for people in the model's susceptible (S), exposed (E), infected (I), and recovered (R) states (cf. Fig. 2).

future waves[3,9]. And finally, the population in China likely had little immune protection against the Omicron variant, as we show below. The epidemic in China provided a unique situation for directly estimating the 'intrinsic' transmissibility of the Omicron variant, which has been a central issue in understanding its evolution[10–12].

The major difficulty in understanding the Omicron epidemic in China is the lack of data directly tracking the spread. Compulsory mass testing and reporting were gradually stopped from Nov. 11 onward, and as we show below, the official numbers of confirmed cases subsequently do not reflect the true extent of SARS-CoV-2 spread. Indirect data must therefore be used. Here, we quantify the Omicron epidemic dynamics in China by drawing on results from online surveys of COVID-19 infection status conducted by a website for the Chinese health authorities on Dec. 26[13]. Despite its limitations (discussed below), the survey represents the currently best evidence of the extent of SARS-CoV-2 spread in China by this time. We developed a modeling approach that integrates both the official case count data before Nov. 11 and the Dec. 26 survey data to reveal a more complete picture of the dynamics of SARS-CoV-2 before and after the dismantling of zero-COVID. Extensive sensitivity analyses confirm that our conclusions are robust to many assumptions about the data and model, and our findings are validated by comparison against two other independent datasets.

## Results

### Understanding the official case count data during the different phases of public health policy
We first aimed to understand the spread of the Omicron variant in China immediately before the relaxation of zero-COVID (i.e., between Oct. 28 and Nov. 11, 2022). Under zero-COVID, testing was mandatory and frequent, leading the official case count data to be a reliable portrayal of the levels of COVID-19 infection in China. Despite the strict measures, such as frequent city-wide lock-downs, travel and movement restrictions, and compulsory quarantine of infected individuals, the data show a clear exponential increase in COVID-19 cases (caused by the extremely transmissible Omicron variants) shortly before the relaxation of zero-COVID (Fig. 1a). Using a negative binomial regression, we estimated the rate of this exponential growth to be 0.14/day (95% CI: [0.13, 0.15]/day) during this period.

Official data after Nov. 11 (when zero-COVID was replaced by 20 Measures and subsequently 10 Measures) report that growth of the SARS-CoV-2 epidemic slowed (Fig. 1a), and eventually the official daily reported cases declined after Nov. 27. This pattern is unexpected. Presumably, the relaxation of strict interventions under zero-COVID would lead to more rapid spread, contrary to what is observed from the official case count data. We reason that the only plausible explanation is the cessation of the requirement for mass testing and rigorous reporting, which led to a rapid decoupling of the surveillance data from the epidemic intensity. As a result, the official case count data after Nov. 11 reflect the rapidly declining detection rate rather than real change in the infection dynamics.

### Estimating SARS-CoV-2 transmission dynamics in Nov. and Dec. 2022
To compensate for the under-reported official case count data after Nov. 11, we collected results from a large online survey conducted on Dec. 26, 2022 by a website for the Chinese Ministry of Human Resources and Social Security[13]. The survey reported the fraction of individuals in three categories: those who thought they had never been infected, thought they currently had COVID-19 symptoms, or thought they had been but were no longer symptomatic (see "Methods" for details). Note that these survey categories do not directly correspond to the epidemiological states of

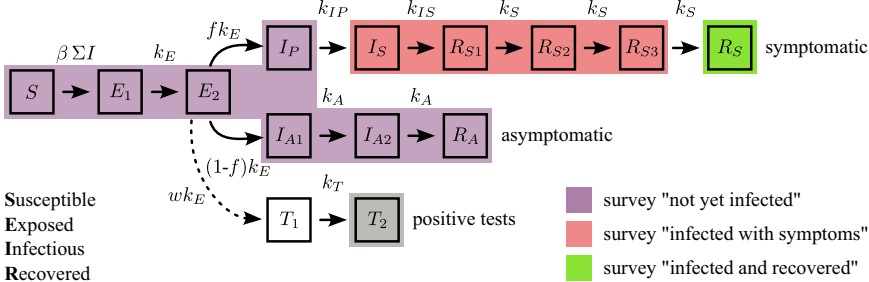

**Fig. 2 | Schematic of the SEIR-type model.** Our model maps the four epidemiological states-susceptible ($S$), exposed ($E$), infectious ($I$), recovered ($R$)-to the three states of the online survey. Individuals who report themselves as infected and recovered (green state) are only those who previously experienced symptoms (top track of states). Those reporting as infected with symptoms (red states) could be currently infectious or recovered but still symptomatic. All others report as not yet infected (purple states), whether they are susceptible, exposed, pre-symptomatic, or asymptomatic. People in all four infectious states can infect susceptibles, with transmission rate $\beta$. A proportion $f$ of infected people eventually develop symptoms. Other parameters are rates of becoming infectious ($k_E$), becoming symptomatic ($k_{IP}$), resolving symptoms ($k_{IS}$, $k_S$), and recovering without symptoms ($k_A$). Some epidemiological states are divided into substates (e.g., $E_1$ and $E_2$) to allow more realistic distributions of waiting times. The bottom track of states counts the fraction $w$ of people who test positive with a delay governed by rate $k_T$; note that this does not remove them from the epidemiological or survey states (thus with a dashed arrow). The corresponding system of ordinary differential equations, along with the rate parameter values, is provided in "Methods".

susceptible, infected, and recovered, because each person is reporting based on symptoms and does not know their true infectiousness status. In particular, individuals may become infectious before their symptoms begin, and their symptoms may linger even after they are no longer infectious. When interpreted correctly, however, these data should provide insight into the epidemic stage and thus allow inference of the epidemic dynamics.

We developed an expanded susceptible-exposed-infected-recovered (SEIR) model that mapped reporting based on symptoms to true infection status. The states of the model are explained in Fig. 2, and equations and further details are provided in "Methods". We separated the model into three time periods to allow changes in official policy to affect the transmission rate (Fig. 1a). Official case counts were reliable for the first period, up to Nov. 11, due to mandatory and frequent testing. There is no clear observation for the second period, so we simply assumed the transmission rate must be no less than in the first period. The dynamics of the third period were then mainly determined by the population survey data on Dec. 26.

Our model assumes no pre-existing population immunity against Omicron. We calculated the levels of vaccine-induced population immunity against Omicron infection based on published vaccination data[14] and the decay of vaccine effectiveness against infection[15] (see "Methods" and Fig. S2). We found that the population immunity before the Dec. 2022 wave of Omicron was below 0.1%, and thus would have negligible effect on the model inference.

Fitting the model to the case count and survey data, we estimated that the outbreak in China grew exponentially at a rate of 0.15/day (95% credibility interval (CrI): [0.13, 0.15]/day) under zero-COVID between Oct. 28 and Nov. 11, increased slightly to 0.17 [0.14, 0.17]/day under 20 Measures, and increased dramatically to 0.42 [0.35, 0.44]/day immediately after full exit from zero-COVID (Fig. 1c), with the epidemic then peaking around Dec. 23 (Fig. 1d). This translates to an epidemic doubling time of 4.7 [4.5, 5.4] days, 4.2 [4.0, 5.0] days, and 1.6 [1.6, 2.0] days for the three periods, respectively.

Assuming a mean generation interval of 3.3 days for Omicron[16] (see "Methods"), we calculated the reproductive number, $R_{exp}$, during the exponential increase stage of the three periods. It increased from 1.57 [1.49, 1.59] during the end of zero-COVID to 3.13 [2.66, 3.27] at the start of the full exit from zero-COVID (Table S1, Fig. 1c). Based on these values, we estimated that the zero-COVID policy prior to relaxation suppressed the transmission of these Omicron variants by 56% [46%, 64%].

Our results suggest that on Dec. 7, the day when full exit from zero-COVID was announced, there were ~1 million new infections.

Because of the extremely high rate of spread afterwards, the outbreak ballooned such that 97% [95%, 99%] of the population (i.e., 1.37 billion people) became infected in December. As a result of the exponential nature of the spread, the vast majority of people (88% [83%, 93%] of the population) became infected during the short window of time between Dec. 15 and 31, 2022 (Fig. 1d).

## Robustness to model and data assumptions
We tested the robustness of our findings—particularly, that nearly the entire population became infected in Dec. 2022—to many properties of the data and model.

First, we previously assumed that the numbers of confirmed cases from Oct. 28 to Nov. 11 represented the actual number of infected individuals, i.e., that the case reporting rate in China was 100%. This is a reasonable approximation because of the mandatory mass testing in place: all individuals were required to get tested every 2–5 days. However, it is possible that some infected individuals were not detected. We therefore considered an alternative in which the reporting rate was only 50%. Here, the estimated growth rate after Dec. 7 was reduced slightly to 0.40/day, the size of the susceptible population by the end of Dec. remained below 4%, and the epidemic again peaked on Dec. 23 (Fig. S3).

Second, we previously assumed the Dec. 26 survey results were an unbiased random sample of the entire population. If, however, recovered people were more likely to respond to the survey, or people reported symptoms not due to COVID-19 (e.g., instead due to influenza—an unlikely scenario given the recent report by China CDC[5]), the numbers of symptomatic or recovered individuals would be over-reported and our growth rate estimate would be too high. We therefore made a large perturbation of the Dec. 26 data by moving 20% of people from each of the symptomatic and recovered categories into the uninfected category. The estimated growth rate was then reduced slightly to 0.39/day after Dec. 7, the size of the susceptible population by Dec. 31 remained below 4%, and the epidemic peak moved slightly to Dec. 24 (Fig. S4).

Third, in our inference of the transmission rates, we fixed many parameter values from the literature. We tested the sensitivity of our results to those assumed values by re-fitting the model using different values. In one strategy, we set the value of each state transition parameter ($k_E$, $k_{IP}$, $k_{IS}$, $k_S$, $k_A$, and $k_T$; all defined in Fig. 2, Table S3, Eq. 1) to either 25% higher or lower than our baseline value (Fig. S5). In another strategy, we assigned an appropriate amount of uncertainty to the incubation period, pre-symptomatic period, symptomatic period, generation interval, and time to PCR testing, drew values from each

distribution, and derived the state transition parameters (details in Table S3); this was repeated 25 times. In all cases, the median exponential growth rate remained above 0.25/day after Dec. 7, the median size of the population still susceptible by Dec. 31 remained <6% and the 95% CrI remained <10%, and the epidemic peak remained between Dec. 21–25.

Fourth, we previously assumed that the entire population is well-mixed. If, however, population structure exists such that people who have high contact rates (and thus potentially more exposure to infection, such as people who live in urban areas or younger age groups) were also more likely to be represented in the data, our results could over-state the total number of infected people by failing to recognize subpopulations that remain uninfected because of their lower contact rates. To test the impact of population structure on our conclusions, we extended our model to build a meta-population model with two subpopulations: one with high contact rates and the other with low contact rates. We assumed that the official case counts and the Dec. 26 survey data were taken from the population with higher contact rates (the sampled population) and the population with lower contact rates (the unsampled population) was not represented in the datasets. We considered various scenarios with different assumptions about the contact rates between the two subpopulations and the contact rate in the unsampled population (see Fig. S6 and "Methods"). In general, we found an epidemic peak on Dec. 23 or Dec. 24, and that >90% of the entire population was infected by the end of Dec. (Fig. S7), consistent with our main findings. This suggests that for a virus that transmits extremely efficiently and rapidly, like Omicron, population structure and heterogeneity play only a small role in dictating the overall rate and magnitude of the epidemic, as long as subpopulations are not entirely isolated from each other.

Overall, allowing for under-reporting of official case count data before Nov. 11, over-reporting of symptoms in the Dec. 26 survey, uncertainty in parameter values taken from the literature, or population sub-structure did not change our conclusions.

**Model validation using additional datasets**

We furthermore validated our results by comparing them against datasets and analyses that were not used in the inference above. First, China CDC subsequently released data on voluntary PCR and antigen tests for most of Dec. 2022 and Jan. 2023[5]. After the full exit from zero-COVID, the test strategy was changed from all population-based to those who request to be tested and groups at risk. The total number of tests declined from a peak of 150 million tests on Dec. 9, to 7.54 million on Jan. 1. The test positivity rate ranged between a few percent to a peak of 29.4% on Dec. 25[5]. Although these data may not directly reveal the magnitude of the true case counts because of the voluntary nature of the reports, they can reveal the pace and the peak timing of the epidemic. We adjusted our model predictions to reflect the nature of voluntary tests (see "Methods"), and found surprisingly close agreement with the national testing data in Dec. 2022 (Fig. 3).

Second, the Sichuan CDC conducted a large daily survey in which residents of the province reported their infection status, the date of test positivity, and the date of symptom onset[17]. We found that the exponential growth rate of the epidemic in early Dec. in Sichuan, as measured by test positivity or symptom onset (0.37/day or 0.43/day, respectively; Fig. S8) is remarkably consistent with our estimates above for the country as a whole.

Third, an analysis of self-reported infection rate focused on vaccine efficacy but also reported an epidemic curve[18]. Based on 2316 survey responses, the study reported that 82.4% of people in China had confirmed cases by Feb. 2023. This number excludes, however, asymptomatic cases that were not revealed by testing. If 9% of infected people never developed symptoms (median value from Fig. S1) and most of those cases were not caught by testing (due to the lack of mandatory testing and the lack of incentive for most non-symptomatic

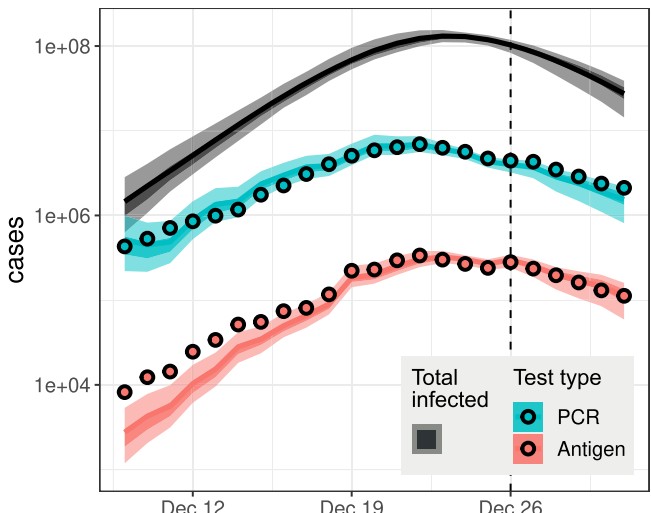

**Fig. 3 | Model comparison with voluntary testing survey from China CDC.** The gray band shows the incidence inferred by our main model fit, which did not include the data points plotted here. Those model predictions adjusted for testing effort and willingness to test (see "Methods") are shown by blue and red bands for the PCR and antigen tests, respectively. Dark and light bands show 50% CrI and 95% CrI derived from the model fit.

people to test themselves, leading to the high test positivity rate by end-Dec[5].), it is likely that more than 90% of people in that survey had been infected. The small sample size of that survey would also induce large uncertainty, leading to a result for the final epidemic size that is not inconsistent with our model-based analysis on larger datasets.

## Discussion

We modeled the epidemic dynamics of the Omicron variant of SARS-CoV-2 in China from Nov. to Dec. 2022, during a period when China moved from having strict COVID-19 policies ('zero-COVID'), to little-to-no intervention efforts. We found that after full exit from zero-COVID, the Omicron variant spread at a very high rate of 0.42/day, with a doubling time of 1.6 days, during early and mid-Dec. before the incidence peaked around Dec. 23. Our point estimate is that 97% of the population (1.4 billion people) was infected during December, with a lower 95% credibility interval of 95% of the population (1.33 billion) and a lower limit in the sensitivity analyses of 90% (1.26 billion). With an infection fatality ratio between 0.1 and 0.2% for the Omicron variant[19], we would expect between 1.3 and 2.6 million COVID-19 deaths in China during Dec. 2022 as well as Jan. 2023 (because of the delay from infection to death).

Infection of more than one billion people during one month would lead to a large number of people needing health care, far exceeding the hospital capacity, and thus explains the report that hospitals were overwhelmed during this period[6]. This outcome emphasizes the need for gradually relaxing intervention efforts (instead of abruptly changing policy) and implementing additional measures (e.g., pharmaceutical strategies) to 'flatten the curve'[9]. A slower epidemic wave would help to ease hospital burden, allow sufficient health care for infected people, prevent epidemic overshoot (i.e., a large final epidemic size because of rapid spread[20]), and ultimately reduce the number of deaths. Such considerations may be of renewed importance later in 2023, when the large cohort of people infected at the end of 2022 may become susceptible to reinfection.

Our results differ from a recent paper by Leung et al.[21] that modeled the epidemic dynamics in Beijing over a similar period. First, they estimated that the highest rate of epidemic growth occurred in mid-Nov., a period when substantial intervention efforts were still in place, whereas our estimates suggest the rate of epidemic growth was

highest after the full exit from zero-COVID. Second, they estimated that the epidemic incidence in Beijing had two peaks (one on Dec. 10 and another on Dec. 21), whereas we estimated the epidemic in China as a whole peaked on Dec. 23, remarkably consistent with the newly released data from China CDC[5] (Fig. 3). These results in Leung et al.[21] are strongly driven by the assumption that the overall contact rate is a function of the daily number of subway travelers, which peaked mid-November. However, it is unclear how the number of subway passengers is related to transmission, given that a large fraction of transmission likely occurs in the household[22]. More generally, it has been shown that human mobility after the first wave of the epidemic in 2020 relates to changes in transmission in complex ways[23]; the relationship between the mobility measures and transmission can be insignificant in many countries, country-specific, or intervention policy-specific. As a result, the results in Leung et al.[21] could be biased by this strong assumption.

One limitation of our analysis is that it is heavily dependent on the online infection status survey data on Dec. 26. We directly addressed two concerns about these data. First, the survey could over-state the progression of the epidemic if people with symptoms are more likely to take the survey, or if people mistake influenza for COVID-19 symptoms. Recent data from the China CDC report[5] shows that among individuals who had influenza-like illnesses and were tested for both COVID-19 and influenza, only a very small fraction were positive for influenza, especially in late Dec. Nevertheless, we artificially increased the proportion of uninfected people in the survey and obtained very similar results. Second, because the survey was conducted online, the demographic characteristics of the participants may be different from the general population. In particular, if a substantial subpopulation did not participate in the survey and also experienced lower rates of spread–for example, people living in sparsely populated areas–our estimate of the total number of people infected could be too high. We tested this assumption by fitting a structured population model and obtained very similar results. Finally, we validated our model findings against other data sources that were not used in our model inference. We found strong concordance with both the estimated peak time of the epidemic and its growth trajectory. These many lines of evidence strongly suggest that our estimation is reliable and robust, despite the limitations of the available data.

Another piece of evidence supporting our conclusion is the lack of a COVID-19 wave in January 2023 (for example, see the China CDC report[5]). The Chinese New Year travel rush, i.e., during which many migrant workers and students go back to hometown to celebrate the Chinese New Year, occurred in Jan., and a large fraction of these travel routes would be from cities to rural villages. Presumably, this travel rush would increase the frequency of risky contact substantially throughout China. The fact that there is no indication of any sizable COVID outbreak during this period (e.g., indicators of COVID-19 infection, including positive tests and fever clinic visits, decreased continuously throughout Jan. 2023[5]) strongly suggests that a large majority of individuals are already infected in Dec., as we predicted in the model.

**Broader implications for understanding the transmission of the Omicron variant and for public health**
Unlike most other countries where there existed sizable levels of population immunity against the Omicron variant arising from recent natural infection and/or vaccination, we estimated that the population in China had very little to no protection against the Omicron variant. Therefore, the growth rate 0.42/day (translating to an epidemic doubling time of 1.6 days) during the exponential growth period we estimated may be a good approximation of the intrinsic transmissibility of the variants BA.5 and BF.7 in a densely populated area without population immunity. This is in stark contrast to the growth rate 0.29/day (and an epidemic doubling time of 2.4 days) that we estimated for the

Wuhan outbreak before intervention efforts were in place in Jan. 2020[24]. This highlights the extremely high intrinsic transmissibility of the Omicron variant in a naive population.

Assuming a mean generation interval of 3.3 days[16], we estimated the reproductive number during exponential growth, $R_{exp}$, to be 3.1. Note that a recent study using contact tracing data from Italy estimated the intrinsic generation time to be 6.8 days[25], although the realized generation time for household transmission where frequent contacts occur between family members was 3.6 days. This relatively large difference in the estimated generation times suggests another layer of complexity in calculating the reproductive number, i.e., it depends on whether the transmission is mostly driven by household transmission (frequent contacts) or non-household transmission (less frequent contacts), which may change in different epidemiological contexts and epidemic stages. The reproductive number, $R_{exp}$, we calculated here is likely accurate when the transmission is dominated by household transmission, e.g., when prevalence is relatively high in a community. However, at the beginning of an epidemic when only a few individuals are infected, the transmission is likely to be a mixture of household and non-household transmission; in this case, the basic reproductive number, $R_0$, is likely to be higher than $R_{exp}$ we estimated here.

Quantifying the growth rates before and after zero-COVID, we estimated these intervention efforts reduced the transmission of the Omicron variant by around 56%. As a comparison, the effect of lockdown during the spread of the original variant in early 2020 was estimated to be 70–80% in Europe[26]. Interestingly, the Omicron outbreak in China had already been growing in Nov. 2022 before the relaxation of zero-COVID (albeit at a low rate). This may reflect the high transmissibilty of the variant, and that the stringent measures of the zero-COVID policy were not effectively implemented because of factors such as population noncompliance due to pandemic fatigue. Nonetheless, these results highlight the difficulty of containing a respiratory infection that causes explosive outbreaks and transmits during the presymptomatic or asymptomatic period.

Because the epidemic grew so rapidly in Dec., we may expect another sizable wave of infection later in 2023 when the large cohort of people infected at the end of 2022 become susceptible to reinfection. The timing and magnitude of the future wave would depend on how quickly the population immunity wanes over time[27–29] and the ability of newly emerging variants to evade the immunity caused by the Omicron variants[30,31]. Therefore, modeling efforts that evaluate intervention strategies, including vaccination, will be crucial to reduce the level of infection and mortality.

## Methods
### Datasets and sources
**Official case counts.** National Health Commission (NHC) of the People's Republic of China had been releasing reports on COVID-19 infections from the beginning of the pandemic until Dec. 23, 2022[32]. These reports include numbers of symptomatic, asymptomatic, imported and recovered infections, and deaths across the country. Because the definitions of 'symptomatic' and 'asymptomatic' infections have been changing over time, we use only the total number of cases by summing the two categories. We retrieved all official data from Sina Pandemic Map[33], which provides pre-processed and well-organized daily case data from NHC. Note the NHC data are also used as a source for the Johns Hopkins COVID-19 Dashboard[34].

**Prevalence survey on Dec. 26, 2022.** On Dec. 26, 2022, the RenShe-Tong online platform (https://m12333.cn), a website for the Chinese Ministry of Human Resources and Social Security, conducted a one-day online survey of the COVID-19 infection status for people living in China[13]. There were 47,897 participants in total. In the questionnaire, participants were asked about their infection status and their duration

of symptoms. Infection status was reported in four categories. 'Uninfected' was defined as individuals who had never tested positive or were unaware they were infected. 'Asymptomatic' was defined as individuals who had tested positive, but did not have symptoms of infection at the time of the survey. 'Symptomatic' was defined as individuals who had tested positive or thought they were infected, and were experiencing symptoms at the time of the survey. 'Recovered' was defined as individuals who had tested positive or thought they had been infected by COVID-19 in the past, and had recovered from symptoms of infection. We interpreted the last category to imply recovered from recent infection, i.e., in end-2022, but there would be little consequence if it also included people who had been infected in previous years because <1% of the population had been infected prior to Nov. 2022[34]. Survey questions are reproduced in Table S2 and survey results by province are provided in a supplemental data file. Because of the small number of respondents in the Asymptomatic category, our analyses folded them in with the Uninfected category, which already included individuals who could be infected but without symptoms. Note that in the survey 95% of infected people reported symptoms, which is consistent with the fraction of infections estimated to be symptomatic by our model (Fig. S1). We computed the total number of people in each survey category under the assumption that the proportion of survey respondents in each category was representative of the respective province as a whole. That is, the prevalences in Fig. 1b are the country's total population size multiplied by 28.3% Uninfected, 42.5% Infected, and 29.2% Recovered.

Participants in the survey were also asked about the duration of symptoms. We fitted an Erlang distribution to the survey data by estimating the mean of the distribution and adjusting the shape parameter of the Erlang distribution from 1 to 10.

**Chinese CDC reports of number of positive tests after Dec. 9, 2022.**
The Chinese Center for Disease Control and Prevention released national COVID-19 pandemic data on Jan. 25, 2023, including PCR test results, antigen test results, outpatient and inpatient data, variant surveillance data, and vaccination progress across the country[5]. Data were collected from official surveillance systems, labs, hospitals, and mobile applications where residents could upload their antigen test results voluntarily. We used daily numbers of positive PCR and antigen tests and daily test positivity rates.

**Sichuan survey on Dec. 25 and 26, 2022.** The Sichuan Center for Disease Control and Prevention Health conducted an online survey of the COVID-19 infection status for people living in Sichuan province on Dec. 24 and 25[17]. There were a total of 158,506 participants. In the questionnaire, participants were asked about their infection status and their duration of symptoms. Infection status was categorized as 'uninfected' or 'tested positive either by PCR or antigen test.' In addition, participants reported the date that they tested positive and the date of their symptom onset.

Copies of all datasets are included as supplemental files.

**The expanded SEIR model**
We developed an SEIR-type model to encompass not only the daily case counts from extensive official testing, but also the survey data in which respondents' answers were based on symptoms. In this model, we categorized the population into 12 classes, plus 2 additional classes for bookkeeping: $S$ for susceptible, $E_1$ and $E_2$ for exposed, $I_P$ for presymptomatic and infectious, $I_S$ for symptomatic and infectious, $R_{S1}$, $R_{S2}$, and $R_{S3}$ for recovered from infectiousness but still symptomatic, $R_S$ for recovered and no longer with symptoms, $I_{A1}$ and $I_{A2}$ for asymptomatic and infectious, and $R_A$ for recovered and never having symptoms. Multiple stages were used for some categories (e.g., stages $E_1$ and $E_2$ for the exposed category) in order to obtain more realistic distributions of waiting times[35], as discussed below. States $T_1$ and $T_2$ were

used to record individuals who test positive, in order to compare with daily case counts. Figure 2 shows how individuals transit from one class to another. Parameters $k_i$ govern the rates of flow from one class to the next, $f$ is the fraction of infected individuals who eventually develop symptoms, and $w$ is the proportion of cases found by testing. The system of equations defining this model is:

$$\frac{dS}{dt} = -\beta(I_P + I_S + I_{A1} + I_{A2})S/N \tag{1a}$$

$$\frac{dE_1}{dt} = \beta(I_P + I_S + I_{A1} + I_{A2})S/N - k_E E_1 \tag{1b}$$

$$\frac{dE_2}{dt} = k_E E_1 - k_E E_2 \tag{1c}$$

$$\frac{dI_P}{dt} = k_E f E_2 - k_{IP} I_P \tag{1d}$$

$$\frac{dI_S}{dt} = k_{IP} I_P - k_{IS} I_S \tag{1e}$$

$$\frac{dR_{S1}}{dt} = k_{IS} I_S - k_S R_{S1} \tag{1f}$$

$$\frac{dR_{S2}}{dt} = k_S R_{S1} - k_S R_{S2} \tag{1g}$$

$$\frac{dR_{S3}}{dt} = k_S R_{S2} - k_S R_{S3} \tag{1h}$$

$$\frac{dR_S}{dt} = k_S R_{S3} \tag{1i}$$

$$\frac{dI_{A1}}{dt} = k_E(1-f)E_2 - k_A I_{A1} \tag{1j}$$

$$\frac{dI_{A2}}{dt} = k_A I_{A1} - k_A I_{A2} \tag{1k}$$

$$\frac{dR_A}{dt} = k_A I_{A2} \tag{1l}$$

$$\frac{dT_1}{dt} = k_E w E_2 - k_T T_1 \tag{1m}$$

$$\frac{dT_2}{dt} = k_T T_1 \tag{1n}$$

In this model, susceptible individuals are infected by infectious individuals (in compartments $I_P + I_S + I_{A1} + I_{A2}$) through contacts at rate $\beta$. Infected individuals first become exposed but not infectious. We used two compartments ($E_1$ and $E_2$) to model this period to reflect the fact that the distribution of the exposed period (i.e., from infection to becoming infectious) usually follows a gamma distribution rather than an exponential distribution. Exposed individuals then become infectious, and $I_P$ and $I_S$ represent infectious individuals at the presymptomatic and symptomatic stages, respectively. Individuals lose infectiousness a few days after symptom onset[36], whereas symptoms may last for a longer period of time. Stages $R_{S1}$, $R_{S2}$, and $R_{S3}$ describe symptomatic individuals who are no longer infectious. Again, using three stages allows us to match the distribution of the symptomatic

period to the distribution estimated from data[13]. Note that, in the survey, participants in the 'infected' category were categorized based on whether the participants have COVID symptoms or not. Therefore, the 'infected' category corresponds to individuals in the states $I_S$, $R_{S1}$, $R_{S2}$, and $R_{S3}$. When individuals recover from symptoms, they enter state $R_S$.

Our model also keeps track of asymptomatically infected individuals, i.e., individuals who never develop symptoms throughout infection. In the model, we assume a fraction, $1 - f$, of individuals remain asymptomatic after the exposed period. We use two $I_A$ stages to describe the duration of infectious period as a gamma distribution. $R_A$ keeps track of individuals recovered from asymptomatic infection. We do not differentiate the stages of individuals recovering from asymptomatic infection (as we did for individuals recovering from symptomatic infection) because individuals in all of these stages would have reported as the same category in the symptoms-based survey, and they remain in this class.

Overall, our model structure allowed us to map our model states to the categories in the Dec. 26 online survey. People who reported themselves 'uninfected' could be truly susceptible ($S$), asymptomatic but exposed ($E_1, E_2$) or infected ($I_{A1}, I_{A2}$), or recovered never having had symptoms ($R_A$). People who reported themselves 'infected' were experiencing symptoms, so they could be still infectious ($I_S$) or no longer infectious ($R_{S1}, R_{S2}, R_{S3}$). People reporting themselves 'recovered' were no longer experiencing symptoms ($R_S$).

To fit the model to official COVID-19 case counts, we used $T_1$ and $T_2$ to keep track of individuals who would eventually be tested. Note that they do not contribute to the transmission dynamics. Parameter $w$ is the fraction of infected individuals who are tested, and it ranges between 0 and 1. The waiting time from becoming infectious to being tested positive is $1/k_T$.

## Parameter values and statistical inference

We fixed most of the parameters in the model to values estimated from epidemiological studies. We used an incubation period for Omicron of mean 3.4 days[37] and a generation interval for Omicron of mean 3.3 days[16]. Both distributions can be approximated by a gamma distribution with a shape parameter of 3, so in our model there are three states describing individuals who are infected but yet to develop symptoms ($E_1$, $E_2$, and $I_P$). We then set the rate parameters $k_E = 2/1.9$ /day and $k_{IP} = 1/1.5$ /day so that the mean pre-symptomatic infectious period is 1.5 days[38] and total incubation period is 3.4 days[37], and $k_{IS} = 1/1.5$ /day so that the mean infectious period after symptom onset is 1.5 days.

We estimated from the population survey data (Table S2) that the distribution of the symptomatic period can be approximated by a gamma distribution with a mean of 5.7 days and a shape parameter of 4[13]. Therefore, in our model, we used four states to represent the symptomatic period ($I_S$, $R_{S1}$, $R_{S2}$, and $R_{S3}$), and we set $k_S = 3/4.2$ /day such that the mean duration that an individual is in the $R_{S1}$, $R_{S2}$ or $R_{S3}$ state is 4.2 days and thus the mean symptomatic period is $4.2 + 1.5 = 5.7$ days. We also set $2/k_A = 1/k_{IP} + 1/k_{IS}$ such that the mean infectious period of the asymptomatically infected is assumed the same as symptomatically infected. Parameter values are summarized in Table S3.

Our model contained six free parameters: the transmission rate during each of the three periods ($\beta_i$), the fraction of infected individuals who eventually develop symptoms ($f$), an initial number of exposed individuals in mid-Oct. ($E_0$), and an overdispersion parameter for the count data ($\phi$), which were assumed to follow a negative binomial distribution.

For the initial condition of the ODE system, we assumed an entirely susceptible population except for $E_1 = E_2 \equiv E_0$ on Oct. 22, one week before the beginning of the case count data we fit. This is consistent with very little pre-existing immunity through either prior

natural infection (<1% of the population had ever been infected by this time[34]) or vaccination (Fig. S2). The model was fit using Bayesian inference and implemented in the Stan programming language[39]. We set the prior distribution of each $\beta_i$ to be normal with mean 3 and standard deviation 2, the prior on $f$ to be uniform between 0 and 1, the prior on $E_0$ to be uniform between 1 and ten times the number of cases on the first day of data, and the prior on $\phi^{-1}$ to be exponential with mean 5. These are weak priors, and the posterior estimates (shown in Fig. S1) were not sensitive to them.

The growth rate, $r$, during the exponential growth period of the outbreak was calculated as the dominant eigenvalue of the Jacobian matrix of the ODE system, Eq. 1, assuming the population is fully susceptible (i.e., $S = N$).

## Calculation of the reproductive number during exponential growth, $R_{exp}$

We used the growth rates, $r$, of SARS-CoV-2 in China estimated in this study and the distribution of the intrinsic generation interval to calculate the reproductive number $R_{exp}$. This is the estimated instantaneous reproductive number during the exponential growth phase of the outbreak after Dec. 8 and before the peak. According to Abbott et al.[16], the distribution of the generation interval has a mean of 3.3 days and a shape parameter close to 3. $R_{exp}$ is then calculated using the formula provided in Park et al.[40]:

$$R_{exp} = \frac{1}{\int g(\tau) \exp(-r\tau) d\tau} \tag{2}$$

where $g$ is the density function of the intrinsic generation interval distribution.

## Regression to estimate exponential growth rates

The rates of exponential growth in Sichuan province during Dec. 2022 were estimated by fitting a linear regression model to the fractions of COVID-19-positive individuals or individuals whose symptoms began between Dec. 2 and Dec. 9. The Dec. 1 data point was ignored because it is likely to include individuals who tested positive or started to have symptoms before Dec. 1. The fitting was performed using the `lm()` function in the R programming environment[41].

We estimated the rate of exponential growth using the official case count data by fitting a generalized negative binomial regression to the case count time series. Fitting was performed with the `glm.nb()` function in the R programming environment[41].

## Estimating population immunity against Omicron infection

Lau et al.[15] estimated vaccine effectiveness (VE) of the CoronaVac vaccine against Omicron infection: 7 days following immunization, the VE for the second and third doses were 5% (0–27%) and 30% (1–66%), respectively, and then the VE reduced exponentially to 1% (0–11%) and 6% (0–29%), respectively, 100 days after immunization. We used these values in our calculation, assuming that all the vaccines administered in China follow the same VE characteristics as CoronaVac. We then collected data on the fractions of people who already received two doses or three doses of vaccine over time from ref. 14 (Fig. S2a). From this dataset, we estimated the fractions of people newly vaccinated with their second or third dose over the entire period of consideration. Note that the fourth dose of vaccine was not authorized in China until mid-Dec. 2022, and therefore we did not include the fourth dose in our calculation.

We calculated the population immunity against Omicron on day $d$, $P(d)$, as

$$P(d) = \sum_{i=d_0}^{d} \sum_{j=2}^{3} G_j(d - i) Q_j(i) \tag{3}$$

where $Q_j(i)$ is the fraction newly vaccinated with the second or the third dose ($j = 2$ or $3$, respectively) on day $i$, and $G_j(d - i)$ is the VE on day $d$ given an individual received the second or the third dose ($j = 2$ or $3$, respectively) on day $i$. The expected and upper bound estimates (Fig. S2b) were made using the expected and upper bound estimates of the VE.

**Robustness of conclusions to existence of population structure**
We considered the impact of population structure on the main conclusions of our model. If population structure exists such that people who have high contact rates (and thus potentially more exposure to infection) are also more likely to be represented in the data, our results could over-state the total number of infected people by failing to recognize subpopulations that remain uninfected because of their lower contact rates. For example, people who live in urban areas with good internet access and health care facilities may be more likely to be included in the COVID-19 testing and online survey, while also experiencing more opportunity for infection; whereas people living in rural areas (far away from testing facilities and with poor internet connections) may not be included in the COVID-19 testing and online survey. Another example is that people in younger age groups may have larger numbers of contacts and be more likely to participate in online surveys than people in older age groups.

To capture this effect as simply as possible, we included two subpopulations: one with high contact rates and the other with low contact rates, with an even lower rate of contact between the two subpopulations. Specifically, we assumed that during each time period, the transmission rate in the low-contact subpopulation was half that of the high-contact subpopulation, and the transmission rate between subpopulations was one-tenth as large. We also assumed that the official case counts and the Dec. 26 survey data were taken from the population with higher contact rates (the sampled population) and the population with lower contact rates (the unsampled population) was not represented in the datasets. The full model structure is provided in Fig. S6.

We first partitioned the entire country's population to be 65% in the high-contact subpopulation and 35% in the low-contact subpopulation. This partition is motivated by the nationwide urban–rural proportion. Because the Dec. 26 survey includes participants from both urban and rural areas, this division represents an overestimate of the size of the unsampled population. Indeed, the majority of population in China has good internet access (through cell phones) and access to COVID-testing sites. Therefore we additionally considered a model where the population is partitioned to be 80% in the high-contact subpopulation and 20% in the low-contact subpopulation, which is probably more appropriate with respect to the survey data. We fit both models to the official case count data and the Dec. 26 survey data as in the main text. Irrespective of the partition, our results from this model were consistent with our main findings. For the 80%–20% and 65%–35% partitions, respectively, we found an epidemic peak on Dec. 23 or Dec. 24, and only 2.6% or 6.7% of the entire population remaining susceptible by the end of December (Fig. S7ab).

We further tested a scenario where the partition is 80% and 20% for the sampled and unsampled subpopulations, respectively, and transmission between the subpopulations was only one-hundredth of the transmission rate within the sampled subpopulation. Note that this is an extreme and unlikely scenario where the two subpopulations were not well connected at all. We found that infection in the low-contact subpopulation lagged behind the high-contact subpopulation even more than in the previous scenarios, as expected (Fig. S7c). The amount of the entire population remaining susceptible by the end of Dec. increased slightly to 10.9%, and the epidemic peak remained on Dec. 23.

**Model adjustment for testing intensity and willingness in the China CDC data**
We compared the fitted model trajectory against the China CDC data to validate our model estimations. For this, we multiplied the total cases predicted by the model by a factor to correct for testing effort and the willingness to get tested. We defined

$$T = C \frac{n_T/p_T}{(N - R)v}, \tag{4}$$

where $T$ is the model-adjusted inferred number of positive tests (plotted as colored bands in Fig. 3), $C$ is the total number of model-predicted cases (gray band in Fig. 3), $n_T$ is the number of positive tests, $p_T$ is the test positivity rate, $N$ is the total country population size, $R$ is the number of individuals in the recovered non-symptomatic model states ($R_S + R_A$), and $v$ is an unknown factor related to the proportion of the population that engages in voluntary testing. The term $(N - R)v$ reflects that not all individuals engage in the testing and among those who engage, individuals who know they are recovered likely do not get tested or self-test any more. Values of all the quantities in Eq. (4) are known for each day—from the model fit or the reported data—except for $v$. We estimated a single value of $v$ (across all days) for each testing type by least-squares fitting, using Eq. (4), obtaining $v = 0.44$ for the PCR tests and $v = 0.52$ for the antigen tests.

**Reporting summary**
Further information on research design is available in the Nature Portfolio Reporting Summary linked to this article.

## Data availability
We collated all data from publicly available data sources, as cited in the main text and Methods. We used WebPlotDigitizer (https://automeris. io/WebPlotDigitizer/) to digitize figures and extract data from figures when numeric values were not available. Cleaned data files are provided at https://github.com/eeg-lanl/cov2-china2022.

## Code availability
Code to reproduce our work is provided at https://github.com/eeg-lanl/cov2-china2022.

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

## Acknowledgements

The research presented in this article was supported by the Laboratory Directed Research and Development program of Los Alamos National Laboratory under project number 20230830ER.

## Author contributions

Study design: R.K. and E.E.G. Data collection and processing: Q.L. and R.K. All authors (E.E.G., Q.L., E.O.R.S., and R.K.) contributed to model construction, data analysis, and writing the manuscript.

## Competing interests

The authors declare no competing interests.
