## [Peer Review File · Nature Communications]

Swift and extensive Omicron outbreak in China after sudden exit from 'zero-COVID' policyREVIEWER COMMENTS

Reviewer #1 (Remarks to the Author):

In this study, the authors built an SEIR model to investigate the transmission dynamics of COVID-19 in China after the sudden exit from intervention strategies. The model considered epidemiological parameters from literature and was fitted to case count and survey data. The authors also provided multiple sensitivity analysis to show the robustness of the model and their findings.

Major comments

1. Given this is a complicated model, I think the authors should spend more time on the methods part to make it more clear.

1) In common SEIR models, each epidemiological status has a single compartment. In this study, E, I and R were divided into multiple stages. What is the benefit of doing this?

2) In the data, recovered was considered as individuals who tested positive or thought to have been infected in the past. It's been 3 years since the emergence of the virus and individuals who were infected more than a year ago are likely to lose immunity and become susceptible again. Is there an estimation of the proportion of recovered but lost immunity? Will this proportion impact the model results?

3) For those reported positive but asymptomatic, the authors considered them as uninfected because of the small population size. Is the asymptomatic proportion consistent with the asymptomatic ratio used in the model? If not, is there participation bias in the survey data? If there's more unreported asymptomatic infection, is it legit to consider them as uninfected at the beginning?

4) The model with low contact rates needs more detail. Why does the low contact group have less compartments? Are β_{11} and β_{22} the same values? If not, how different are they?

5) Need a table that summarizes all parameter values, both inferred or from literature. This helps the reader to better understand the model.

2. Figures need improvements in general

1) Figure captions should be very clear. Some figure captions use "Figure components are as in Fig XX" and cite previous figures. I was going back and forth to check the reference figure to find out what the labels are. I think it is necessary to describe everything in the caption.

2) Figure format - some legend and labels have capital letters and some do not. Please use the same format for all.

3. Results and discussion

1) The inferred reproduction number R: is this R_0 or R_t ?

2) I think more discussion on test report rate and test sensitivity is needed as this affects the model results and is critical in testing-based intervention strategies.

Minor comments

1. Need to explain what CrI is short for in the first use.

2. Line 5: at a rate of 0.42/day. I was trying to figure out what rate this is - is it 0.42x more infected or anything else? After reading the Results section I found it to be the parameter of the distribution. I think it needs to be explained better in the Abstract.

3. Figure 1a legend: unmodeled is misleading, probably change to not used in model

Reviewer #2 (Remarks to the Author):

In the current manuscript, authors investigate the potential trajectory of Covid-19 epidemic in China after the abrupt stop to Covid-zero policy and lack of regular thereafter. To do so, they employ an SEIR-type mathematical model along with data from an online survey conducted by Chinese Ministry of Human Resources and Social Security in late December. Under several varying reasonable assumptions regarding the survey data, they found that more than 95% of population would have been infected with Omicron variant by the end of December. Authors approach to

utilize the limited data to gain broader insight is noteworthy. However, I am unsure whether it is conclusive. My detailed comments are as follows:

1. Authors use most parameters from literature. However, they only use their point estimates from literature, in a study that is using limited information. Uncertainty in these epidemiological parameters could have considerable impact on the range of their final estimate. Therefore, the current study should take a likelihood approach to the model fitting, incorporate parameter distributions from literature and generate confidence intervals for their estimates.
2. While author employ a meta-population model to consider the heterogeneity in contact mixing, it still may not capture the spatial heterogeneity that may exist and is overlooked when looked at the whole country as a well-mixed population. Authors choice of 65% highly connected vs 35% less connected is a reasonable assumption for the meta-population model approach, but it may not be sufficient for justification. Authors should discuss about how the contact patterns changed with respect to switching of the policies. Moreover, limitation of not incorporating contact patterns in their model explicitly.
3. Please comment on role of pre-symptomatic transmission that is associated with Covid-19. Model assumes only symptomatic individuals transmit.
4. Why was the proportion of infected individuals who develop symptoms fitted? What was the prior used for it? Studies have shown that percentage of asymptomatic Omicron-infected individuals may range from 10-60%. These estimates may be from populations that are highly vaccinated, how does it compare with China? Figure S2 suggests the estimate in China for this proportion to be much higher.
5. Please provide details of the policy shift for readers who do not know what constituted zero covid policy to 20 measures to 10 measures

Reviewer #3 (Remarks to the Author):

Thank you for the opportunity to review the manuscript by Goldberg et al. By developing an expanded SEIR model and fitting it to both the official case count data before Nov. 11 and the Dec. 26 survey data, the authors reconstructed the transmission dynamics of Omicron in China before and after the optimization of prevention and control measures. The analyses are interesting.

However, I have two major concerns. The authors estimated that 97% of Chinese population was infected during December 2022. I wonder whether such a high proportion of the population infected in little more than a month is plausible. As discussed in the limitation paragraph, the proportion is determined by the online prevalence survey data on Dec 26, 2022. The authors are also aware of the bias in the online infection status survey data, and conducted extensive sensitivity analyses. But I am worried that reporting 97% of the entire population (>1.4 billion) infected in the main analysis is less convincing. The same issue also occurred in Leung et al. *Nature Medicine*, 2023, which fit their model to the online infection status survey data and estimated the cumulative infection attack rate (IAR) in Beijing was 92.3% on Jan 31, 2023. It's believed that the cumulative IAR may be overestimated. As a mega city, the population is mixed better in Beijing than in the other cities of China. It's more likely that the cumulative IAR in other cities would be less than 90% considering the geographical heterogeneity in such a large country as China. Moreover, the Omicron outbreak occurred earlier in Beijing, providing a warning for other cities where their citizens could improve personal protection. These changes in population behaviors may further reduce the cumulative IAR in those cities. Thus, the overall IAR in China is unlikely to reach such a high figure (97%).

Another concern is about the epidemic trajectory during the "10 Measures" period (i.e., from Dec 8 to Dec 26, 2022). The authors used a fixed transmission rate β_3 , and the estimated peak time and growth trajectory matched the China CDC data. However, as shown in the China CDC data,

the epidemic trajectory during this period was apparently right skewed, suggesting that using different β s before and after the peak time may be more appropriate.

Minor comments

1. In the expanded SEIR model, it's rare to have states for symptomatic but no longer infectious. Could the authors provide an explanation? Moreover, why use multiple states in the model for a class (e.g., RS1, RS2, and RS3; IA1 and IA2; T1, and T2)?

2. Scientific research should use accurate and objective language. Please rephrase following words.

Page 1 Line 3, "undisclosed".

Page 2 Line 54, "frequent citywide lockdowns".

Page 5 Line 151, there are two "the".

3. The authors estimated that the reproduction number R from the three time periods are all less than 4. However, several studies estimated that the basic reproduction number R0 for Omicron variants was greater than 7. Could the authors discuss the difference in these estimates?

Perez-Guzman PN, Knock E, Imai N, et al. Epidemiological drivers of transmissibility and severity of SARS-CoV-2 in England. medRxiv 2023: 2023.02.10.23285516.

Leung K, Leung GM, Wu J. Modelling the adjustment of COVID-19 response and exit from dynamic zero-COVID in China. medRxiv 2022: 2022.12.14.22283460.

Texts in *italics* are comments from reviewers; texts in normal font are author responses.

Reviewer #1

[1] In this study, the authors built an SEIR model to investigate the transmission dynamics of COVID-19 in China after the sudden exit from intervention strategies. The model considered epidemiological parameters from literature and was fitted to case count and survey data. The authors also provided multiple sensitivity analysis to show the robustness of the model and their findings.

1. Given this is a complicated model, I think the authors should spend more time on the methods part to make it more clear.

We have endeavored to explain more of the model earlier in the article, primarily within the revised model schematic figure which now appears in the main text (Fig. 2). We also added detailed descriptions of the rationale for the choice of model framework and each of the model terms in the Methods section.

[2] 1) In common SEIR models, each epidemiological status has a single compartment. In this study, E, I and R were divided into multiple stages. What is the benefit of doing this?

When a single compartment is used, as in the traditional SEIR model, the time spent in each epidemiological category (e.g., the latent period modeled by E , or the infectious period modeled by I) becomes exponentially distributed. However, in reality, empirical epidemiological studies have shown that these waiting times are better described by gamma distributions. As pointed out by Wearing et al. [PLoS Medicine 2(8): e320], using the correct distributions is critically important to estimate epidemiological parameters. We therefore employed the common mathematical technique (also described in that paper) of dividing compartments into multiple stages in order to obtain gamma-distributed waiting times. We added this explanation to the Methods section, in the text before and after Eq. 1.

[3] 2) In the data, recovered was considered as individuals who tested positive or thought to have been infected in the past. It's been 3 years since the emergence of the virus and individuals who were infected more than a year ago are likely to lose immunity and become susceptible again. Is there an estimation of the proportion of recovered but lost immunity? Will this proportion impact the model results?

As the initial condition of our model, we assumed that everyone is susceptible at the beginning of the model time period (22 Oct 2022) except for the few people placed in the exposed category. That is, we effectively assumed the complete loss of any immunity gained through natural infection prior to that date. Consequently, the recovered category in the model only includes people who were infected after 22 Oct 2022.

One additional clarification is that the Dec 26 survey asks about recent infection, i.e., within the end-2022 surge, not in previous years. If, however, earlier-infected people reported themselves as recovered, this would have very little effect. China implemented various versions of the zero-COVID policy before Nov 2022 to keep the spread of SARS-CoV-2 at an extremely low level. Earlier-infected people represents a negligible fraction of the population.

In the manuscript, we added these clarifications in the Methods sections titled 'Prevalence survey on Dec. 26, 2022' and 'Parameter values and statistical inference'.

[4] 3) For those reported positive but asymptomatic, the authors considered them as uninfected because of the small population size. Is the asymptomatic proportion consistent with the asymptomatic ratio used in the model? If not, is there participation bias in the survey data? If there's more unreported asymptomatic infection, is it legit to consider them as uninfected at the beginning?

In the Dec 26 survey responses, of the people who reported as currently infected, 95% reported as symptomatic (Table S2). This is consistent with our model estimate of the symptomatic fraction (f in Fig. S1), which had a median

value of 0.91 (95% CrI [0.81, 1.00]). We added this consistency check in the Methods section ‘Prevalence survey on Dec. 26, 2022’.

The other point to note is that the ‘uninfected’ category in the survey data corresponds to several states in our model (Fig. 2). By considering those reported positive but asymptomatic as ‘uninfected’, we are not assigning them as ‘susceptible’ in the SIER model. They are more likely to be pre-symptomatic or asymptomatic individuals (which are also included in the ‘uninfected’ category).

[5] 4) The model with low contact rates needs more detail. Why does the low contact group have less compartments? Are beta11 and beta22 the same values? If not, how different are they?

We improved Fig. S6 to answer both these questions. It is not necessary to model symptomatically and asymptotically infected people separately in the low-contact subpopulation because those people were not measured by the survey (under the assumptions of this meta-population model); we thus kept track of all infected individuals in a single class for simplicity. And, we assume that the transmission rate in the low-contact subpopulation is half that of the high-contact subpopulation.

[6] 5) Need a table that summarizes all parameter values, both inferred or from literature. This helps the reader to better understand the model.

We added this table (Table S3).

[7] 2. Figures need improvements in general

1) Figure captions should be very clear. Some figure captions use “Figure components are as in Fig XX” and cite previous figures. I was going back and forth to check the reference figure to find out what the labels are. I think it is necessary to describe everything in the caption.

We added more information to those figure captions. (Though we also kept the notes about components being the same, so that the reader can skip over the extra caption text if desired.)

[8] 2) Figure format - some legend and labels have capital letters and some do not. Please use the same format for all.

In all figures that have labeled panels, the labels are now lowercase letters.

[9] 3. Results and discussion

1) The inferred reproduction number R : is this R_0 or R_t ?

The reported R values are the estimated instantaneous reproductive number during the exponential growth phase of the outbreak after Dec. 8 and before the peak of the outbreak.

We now refer to this parameter as R_{exp} , for clarity, and its definition is provided in the Methods section ‘Calculation of the reproductive number during exponential growth, R_{exp} ’.

[10] 2) I think more discussion on test report rate and test sensitivity is needed as this affects the model results and is critical in testing-based intervention strategies.

We added additional information on how testing effort changed over time to the ‘Model validation using additional datasets’ subsection. If by ‘test sensitivity’ the reviewer refers to the sensitivity of PCR versus antigen tests, i.e., how sensitively each detects SARS-CoV-2 infection, we feel that properly considering the sensitivity of the tests requires many additional considerations and assumptions (see a work by one of us, Ke et al. PNAS 118 (49) e2111477118, for example), and thus it is beyond the scope of this work, which is modeling focusing more on transmission patterns.

[11] 1. Need to explain what CrI is short for in the first use.

Fixed.

[12] 2. Line 5: at a rate of 0.42/day. I was trying to figure out what rate this is - is it 0.42x more infected or anything else? After reading the Results section I found it to be the parameter of the distribution. I think it needs to be explained better in the Abstract.

This is the rate of exponential increase—a measure of how quickly an epidemic spreads. If we assume the epidemic grows exponentially (as most epidemics do initially), i.e., $I(t) = I_0 e^{rt}$, the reported rate is r . Another way to understand it is that the doubling time of the epidemic is calculated as $\log(2)/r$. We now provide this the perhaps-more-intuitive doubling time in the Abstract as well as Results.

[13] 3. Figure 1a legend: unmodeled is misleading, probably change to not used in model

Fixed.

Reviewer #2

[14] In the current manuscript, authors investigate the potential trajectory of Covid-19 epidemic in China after the abrupt stop to Covid-zero policy and lack of regular thereafter. To do so, they employ an SEIR-type mathematical model along with data from an online survey conducted by Chinese Ministry of Human Resources and Social Security in late December. Under several varying reasonable assumptions regarding the survey data, they found that more than 95% of population would have been infected with Omicron variant by the end of December. Authors approach to utilize the limited data to gain broader insight is noteworthy. However, I am unsure whether it is conclusive.

Thank you for the following additional suggestions to help us draw as-firm-as-possible conclusions from such limited data.

[15] 1. Authors use most parameters from literature. However, they only use their point estimates from literature, in a study that is using limited information. Uncertainty in these epidemiological parameters could have considerable impact on the range of their final estimate. Therefore, the current study should take an likelihood approach to the model fitting, incorporate parameter distributions from literature and generate confidence intervals for their estimates.

We certainly agree that there is substantial uncertainty in the parameter values we used, and we do indeed report credibility intervals for the estimated quantities obtained from our model (Figs. 1 and 3, Figs. S1, S3, S4 and S7, and Table S1, and throughout the text).

To allow for uncertainty in the input parameter values we relied on, we previously adjusted each one up or down by 25% (Fig. S5). In response to this reviewer comment, we added a new means of incorporating parameter uncertainty (Table S3). Now, we (1) report reasonable uncertainties for the incubation period, pre-symptomatic period, symptomatic period, generation interval, and time to PCR test result, (2) draw values from those distributions, (3) obtain the transition rate parameters from those values, and (4) fit the model with those new transition rate parameters. Our results remain unchanged, as reported in the ‘Third’ point of the Results section ‘Robustness to model and data assumptions’.

[16] 2. While author employ a meta-population model to consider the heterogeneity in contact mixing, it still may not capture the spatial heterogeneity that may exist and is overlooked when looked at the whole country as a well-mixed population. Authors choice of 65% highly connected vs 35% less connected is a reasonable assumption for the meta-population model approach, but it may not be sufficient for justification. Authors should discuss about how the contact patterns changed with respect to switching of the policies. Moreover, limitation of not incorporating contact patterns in their model explicitly.

We have enhanced our discussion of the meta-population model in the Results section ‘Robustness to model and data assumptions’, the Methods section ‘Robustness of conclusions to existence of population structure’, and Fig. S6.

A spatially-explicit model is far beyond the scope of the present work, but the simplified contact patterns we do include (differences within and between two subpopulations) illustrate that even extreme population subdivision does not greatly reduce the total extent of infection. By testing various assumptions, we found a general pattern/observation is that because of the high transmissibility of the Omicron variant, a large fraction of the population (>90%) as a whole will be infected, even when the contacts between different subpopulations are infrequent. We expect this will be true even if more complicated contact structure or spatial heterogeneity were considered. We also would like to emphasize that China has a high population density, and even in villages, the population density is relatively high compared to many other parts of the world. Therefore, even within the rural subpopulation (as considered in our model), we expect the contacts will not be very low. Therefore, we would expect the virus would eventually spread very widely in China and infect most individuals in the absence of mitigation efforts.

In terms of changes of contact patterns due to changes in policies, we would expect the contacts within each subpopulation and between them to increase as restrictions are lifted. Our model does incorporate this: the overall transmission rate increases with time (Fig. 1c, Fig. S1), and the transmission rates within and between subpopulations are scaled accordingly. We now explain this in Fig. S6.

[17] 3. Please comment on role of pre-symptomatic transmission that is associated with Covid-19. Model assumes only symptomatic individuals transmit.

Our model does actually allow asymptomatic and pre-symptomatic individuals to transmit the virus, as well as symptomatic infectious individuals. To explain this better, we revised the model schematic figure and legend (now Fig. 2).

[18] 4. Why was the proportion of infected individuals who develop symptoms fitted? What was the prior used for it? Studies have shown that percentage of asymptomatic Omicron-infected individuals may range from 10-60%. These estimates may be from populations that are highly vaccinated, how does it compare with China? Figure S2 suggests the estimate in China for this proportion to be much higher.

Beginning with an estimate of the symptomatic proportion from the literature would have been possible, but as noted by the reviewer, there is much uncertainty in this estimate and it may differ among populations. We therefore chose to provide no information about this quantity—we used a flat prior ranging from 0 to 1, as noted in the Methods section ‘Parameter values and statistical inference’—and to let the model estimate it from the data. The estimated value we obtained is consistent with the survey results (see Point [4] above).

[19] 5. Please provide details of the policy shift for readers who do not know what constituted zero covid policy to 20 measures to 10 measures

We added the description of the policy shift in the Introduction.

Reviewer #3

[20] Thank you for the opportunity to review the manuscript by Goldberg et al. By developing an expanded SEIR model and fitting it to both the official case count data before Nov. 11 and the Dec. 26 survey data, the authors reconstructed the transmission dynamics of Omicron in China before and after the optimization of prevention and control measures. The analyses are interesting.

However, I have two major concerns. The authors estimated that 97% of Chinese population was infected during December 2022. I wonder whether such a high proportion of the population infected in little more than a month is plausible. As discussed in the limitation paragraph, the proportion is determined by the online prevalence survey data on Dec 26, 2022. The authors are also aware of the bias in the online infection status survey data, and conducted extensive sensitivity analyses. But I am worried that reporting 97% of the entire population (>1.4 billion) infected in the main analysis is less convincing. The same

issue also occurred in Leung et al. Nature Medicine, 2023, which fit their model to the online infection status survey data and estimated the cumulative infection attack rate (IAR) in Beijing was 92.3% on Jan 31, 2023. It's believed that the cumulative IAR may be overestimated. As a mega city, the population is mixed better in Beijing than in the other cities of China. It's more likely that the cumulative IAR in other cities would be less than 90% considering the geographical heterogeneity in such a large country as China. Moreover, the Omicron outbreak occurred earlier in Beijing, providing a warning for other cities where their citizens could improve personal protection. These changes in population behaviors may further reduce the cumulative IAR in those cities. Thus, the overall IAR in China is unlikely to reach such a high figure (97%).

We agree that the conclusion of 97% of the entire population (>1.4 billion people) infected is bold. We have addressed this concern in two ways. First, in the Abstract and the beginning of the Discussion, we temper this number by also providing the slightly-lower values from the 95% credibility interval and the various sensitivity checks. Second, we have added content to the Discussion that further supports our finding of such a large December wave of infections.

The strongest evidence supporting our conclusion is that there is no evidence for another COVID-19 wave in January 2023 (for example, see the China CDC report [5]). The Chinese New Year travel rush, during which many migrant workers and students go back to their hometowns to celebrate the Chinese New Year, occurred in Jan., and a large fraction of these travel routes would be from cities to rural villages. Presumably, this travel rush would increase the frequency of risky contact substantially throughout China. The fact that there is no indication of any sizable COVID outbreak during this period strongly suggests that most individuals were already infected in Dec., as we predicted in the model.

We understand predicting 97% of 1.4 billion people were infected in a short period of time is shocking intuitively. However, from a theoretical point view, we estimated that there were around 1 million people already infected at the beginning of Dec. 2022. Because of the exponential nature of epidemics, an epidemic growing from 1 million infected people to a billion people takes the same time for the same epidemic growing from 1 infected to 1000 infected. Therefore, the prediction is mostly because of the rapid and unmitigated spread of Omicron, and a large number of individuals were already infected in early Dec. We include these points in the Discussion.

[21] Another concern is about the epidemic trajectory during the “10 Measures” period (i.e., from Dec 8 to Dec 26, 2022). The authors used a fixed transmission rate β_3 , and the estimated peak time and growth trajectory matched the China CDC data. However, as shown in the China CDC data, the epidemic trajectory during this period was apparently right skewed, suggesting that using different β s before and after the peak time may be more appropriate.

We see the point that the data in Fig. 3 are not symmetric about the peak. A changing value of β is not needed to explain this, however. In SEIR dynamics with fixed β , the rise in infections is steep because of the near-exponential growth within the mostly-susceptible population. The decline in infections is then less steep because the fraction of susceptible individuals gets reduced as many of them are already infected. Given this intuitive explanation, and the lack of sufficient data to estimate a changing value of β during this time period, we think we are justified in estimating a single value for β_3 .

[22] Minor comments

1. In the expanded SEIR model, it's rare to have states for symptomatic but no longer infectious. Could the authors provide an explanation? Moreover, why use multiple states in the model for a class (e.g., RS1, RS2, and RS3; IA1 and IA2; T1, and T2)?

We include states for symptomatic but no longer infectious in order to align the model with the ‘infected’ category in the population survey on Dec. 26. This is now better explained in Fig. 2, Results section ‘Estimating SARS-CoV-2 transmission dynamics in Nov. and Dec. 2022’, and Methods sections ‘Prevalence survey on Dec. 26, 2022’ and ‘The expanded SEIR model’.

Regarding the use of multiple model states within an epidemiological class, please see our response to Point [2].

[23] 2. *Scientific research should use accurate and objective language. Please rephrase following words.*
Page 1 Line 3, “undisclosed”.
Page 2 Line 54, “frequent citywide lockdowns”.
Page 5 Line 151, there are two “the”.

All fixed.

[24] 3. *The authors estimated that the reproduction number R from the three time periods are all less than 4. However, several studies estimated that the basic reproduction number R_0 for Omicron variants was greater than 7. Could the authors discuss the difference in these estimates?*

The estimation of reproductive number is complicated by several factors. Most studies that estimated R_0 greater than 7 for the Omicron variant were based on the relative fitness of Omicron. For example, if R_0 for the Delta variant has been estimated to be 5 and the Omicron variant is 1.5 times fitter than the Delta, the R_0 for the Omicron variant is then around $5 \times 1.5 = 7.5$. This argument assumes that the rate of relative growth observed in data (Omicron vs. Delta in this case) is mostly driven by intrinsic transmissibility of the variant. However, in the case of Omicron, the relative growth is likely to be driven by both higher intrinsic transmissibility and the immune escape (i.e., Omicron is better at infecting individuals previously infected by Delta). Therefore, the calculation of $5 \times 1.5 = 7.5$ is inaccurate, especially for Omicron.

The relatively low reproductive number we estimated for Omicron is also driven by the short generation interval. For a given estimated epidemic growth rate (0.42/day as we estimated for the Omicron spread in China), the shorter the generation interval, the lower the estimated reproductive number. As reported in [16], the generation interval is around 3.3 days for Omicron; this suggests most transmission occurs early in infection. This is in stark contrast to the estimated generation interval for the original variant which is in the range of 5–9 days.

We now describe this issue in the Discussion.

REVIEWER COMMENTS

Reviewer #1 (Remarks to the Author):

The authors have addressed all my questions and concerns in the revision.

Reviewer #2 (Remarks to the Author):

Authors have addressed my comments sufficiently.

Reviewer #3 (Remarks to the Author):

The authors have substantially resolved my issues except for the high cumulative IAR. To address this concern, the authors have reported the sensitivity analysis lower limit of 90% in the Abstract and added empirical evidence to the Discussion. I agree that the absence of another wave in Jan 2023 can be attributed to that most individuals had been already infected in Dec 2022. However, this fact cannot support such a high proportion (97%) of the Chinese population infected. A peer-reviewed study recently published in China CDC weekly estimated only 82.4% of the Chinese population infected as of Feb 7, 2023, which also based on the self-reported infection rate determined from a four-round online survey. Fu et al. also constructed the epidemic curve from Dec 1, 2022 to Feb 5, 2023, which is consistent with the China CDC report. Thus, we can infer that around 85% of Chinese population infected in Dec 2022 would also prevent the occurrence of another wave in Jan 2023.

As pointed out previously, the high cumulative IAR is determined by the Dec 26 survey. Compared to the four-round online survey conducted by Fu et al., the Prevalence survey conducted by the RenSheTong on Dec 26, 2022 is considered less reliable due to its absence of specific details, a public health background, and peer-reviewed status. If the authors were to calibrate their model using the survey data from Fu et al., it is anticipated that a lower estimate of the cumulative IAR could be obtained. Therefore, the conclusion of 97% of the Chinese population infected is not as compelling.

Fu D, He G, Li H, et al. Effectiveness of COVID-19 Vaccination Against SARS-CoV-2 Omicron Variant Infection and Symptoms — China, December 2022–February 2023. *China CDC Weekly* 2023; 5(17): 369-73.

Minor comments

1. References [1] and [2], change “the New Coronary Pneumonia” to “COVID-19”. Please check the translation of Chinese terms.

2. Lines 360-361, Rexp is calculated using the distribution of the intrinsic generation interval with a mean of 3.3 days, the reference for which is not peer-reviewed. Instead, the following peer-reviewed study estimated the mean intrinsic generation time of 6.84 days for the Omicron variant, and a mean realized household generation time of 3.59 days. The value used by the authors is close to the latter.

Manica M, De Bellis A, Guzzetta G, et al. Intrinsic generation time of the SARS-CoV-2 Omicron variant: An observational study of household transmission. *The Lancet Regional Health – Europe* 2022; 19.

3. Line 163, change “having the world’s strictest COVID-19 policies (‘zero-COVID’)” to “the strict ‘zero-COVID’ policy”. As suggested previously, please use accurate and objective language.

4. Line 225, 202 should be 2020.

Reviewer comments are in *italics*, and author responses are in normal font.

Reviewer #1

The authors have addressed all my questions and concerns in the revision.

Reviewer #2

Authors have addressed my comments sufficiently.

Reviewer #3

The authors have substantially resolved my issues except for the high cumulative IAR. To address this concern, the authors have reported the sensitivity analysis lower limit of 90% in the Abstract and added empirical evidence to the Discussion. I agree that the absence of another wave in Jan 2023 can be attributed to that most individuals had been already infected in Dec 2022. However, this fact cannot support such a high proportion (97%) of the Chinese population infected. A peer-reviewed study recently published in China CDC weekly estimated only 82.4% of the Chinese population infected as of Feb 7, 2023, which also based on the self-reported infection rate determined from a four-round online survey. Fu et al. also constructed the epidemic curve from Dec 1, 2022 to Feb 5, 2023, which is consistent with the China CDC report. Thus, we can infer that around 85% of Chinese population infected in Dec 2022 would also prevent the occurrence of another wave in Jan 2023.

As pointed out previously, the high cumulative IAR is determined by the Dec 26 survey. Compared to the four-round online survey conducted by Fu et al., the Prevalence survey conducted by the RenSheTong on Dec 26, 2022 is considered less reliable due to its absence of specific details, a public health background, and peer-reviewed status. If the authors were to calibrate their model using the survey data from Fu et al., it is anticipated that a lower estimate of the cumulative IAR could be obtained. Therefore, the conclusion of 97% of the Chinese population infected is not as compelling.

Fu D, He G, Li H, et al. Effectiveness of COVID-19 Vaccination Against SARS-CoV-2 Omicron Variant Infection and Symptoms – China, December 2022–February 2023. China CDC Weekly 2023; 5(17): 369-73.

Thank you for bringing this new paper to our attention. We read it very carefully, and our team discussed extensively how to include their data in our analysis and/or compare their overall findings with ours. After carefully reading the paper and performing extensive searches using multiple English-language and Chinese-language based search engines, we do not find the the full dataset, and thus are not able to analyze the dataset. Instead, we carefully interpreted the data set and reach the conclusion that the dataset is actually consistent with our findings after correction for biases. We thus revised our manuscript to discuss our findings in light of the paper and the dataset. Below are our responses and explanations regarding the three main points in your comment above:

First, the Fu et al. finding of $\sim 85\%$ of the population infected in Dec 2022 logically must be an underestimate of the true value (although this underestimate may not be critical for the purpose of estimating vaccine efficacy, i.e. the main aim of the Fu et al. study). Their definition of a ‘confirmed case’ is someone who tested positive and/or experienced symptoms. It therefore misses people who were infected but asymptomatic and did not test. The total testing effort among survey participants is not reported, but it is reasonable to expect than many of the asymptomatic cases were missed, given the cessation of mandatory testing during this time and the absence of an incentive for most non-symptomatic people to test themselves. Fu et al. formed their epidemic curve by simply counting up confirmed cases, whereas our model-based analysis allowed for asymptomatic infections (as well as incorporated earlier case-count data and epidemiological dynamics). It is unfortunately impossible for us to analyze their unreported data (see our third point below), but our rough estimate is that at least 10% of cases were missed in that paper (we explain our reasoning in a new section of our manuscript). With this correction, their finding becomes consistent with ours.

Second, the survey data used by Fu et al. are not more reliable than the RenSheTong survey data we used. Of course all surveys have their own flaws. However, comparing these two surveys:

- The RenSheTong survey reported the actual survey questions and the number of responses for each answer (Table S2 in our manuscript). The Fu et al. team conducted the survey themselves and did not report the questions or raw response data. We searched extensively online for any other information about this survey (including the efforts of our Chinese-reading co-authors) and could find no details. The RenSheTong survey is therefore more transparent and interpretable.
- Our analysis used the responses of all 47,897 people who answered the RenSheTong survey. The Fu et al. survey had 10,439 total responses, but only the responses from the fourth round—2,316 people—were used to compute their epidemic curve. The RenSheTong survey therefore provides 20 times more data.
- In analyzing the RenSheTong survey data, we weighted the responses per province by the total population of each province (our supplemental file `survey.csv`). The Fu et al. analysis did not make any geographic corrections, and additionally more than half the responses were from Guangdong province (where the authors are all based) and Zhejiang province, which have better economic and public health conditions than much of the country. The RenSheTong survey analysis is therefore more geographically representative of the entire country.

Third, the necessary survey data were not actually published, so it is not possible for us to analyze them. In order to estimate the epidemic curve (including asymptomatic cases not found through testing) from these survey data, we would need the number of confirmed cases, number of respondents without confirmed cases, and testing effort, to use along with an estimate of the proportion of cases that are asymptomatic. These data are simply not reported in the paper, neither the main text/figures nor the supplementary materials. Note particularly that the data tables have no date information and do not have numbers approaching 85% infected, and the timeseries figures have no testing effort or sample size information. The main focus of the paper was vaccine efficacy, so the data in e.g. Tables S1 & S2 are for the matched set of Case and Control respondents.

Therefore, it is not possible for us to use these new survey data in our model (which does allow for undetected and asymptomatic infections). The best we can do is to use our model results to provide a rough improvement to the total epidemic size reported by Fu et al. We have added this to our Results section ‘Model validation using additional datasets.’ This additional comparison does strengthen our paper, and we appreciate your calling our attention to this recent other publication.

As authors, our team normally makes every effort to respond positively to reviewer comments and use them to improve our work (as you have seen in the other aspects of our previous extensive revision). But although this new Fu et al. paper initially seemed very promising and relevant, a close examination reveals that their epidemic magnitude must be an under-estimate, and that they do not publish the data necessary for us to obtain an improved estimate.

References [1] and [2], change “the New Coronary Pneumonia” to “COVID-19”. Please check the translation of Chinese terms.

Fixed.

Lines 360-361, Rexp is calculated using the distribution of the intrinsic generation interval with a mean of 3.3 days, the reference for which is not peer-reviewed. Instead, the following peer-reviewed study estimated the mean intrinsic generation time of 6.84 days for the Omicron variant, and a mean realized household generation time of 3.59 days. The value used by the authors is close to the latter.

Manica M, De Bellis A, Guzzetta G, et al. Intrinsic generation time of the SARS-CoV-2 Omicron variant: An observational study of household transmission. The Lancet Regional Health – Europe 2022; 19.

Thank you for pointing out this very relevant paper. This study on a large number of infected individuals clearly shows the intrinsic generation time is longer (6.8 days), whereas the realized generation for household transmission is likely to be much shorter (due to frequent contacts and competition for susceptibles within a household). Based on this result, we agree that the *basic* reproductive number is likely to be higher than the reproductive number we calculated, because in the beginning of the epidemic, the transmission is driven by both household and non-household contacts (leading to a longer generation interval than we used here). However, as the epidemic goes on and infection becomes

wide-spread, we would expect the transmission to be largely driven by household transmission, as shown in several epidemiological studies. The “instantaneous” reproductive number during exponential growth should be close to the R_{exp} we estimated.

In the Discussion, we removed the speculation that the Omicron had a similar *basic* reproductive number as the original Wuhan strain, and we added a paragraph discussing the effects of transmission mode on the generation time and reproductive number.

A related note regarding the fraction of total infected individuals: given the high *basic* reproductive number as the reviewer suggested and the high rate of spread, it is expected a large fraction of the population (>90%) would be infected in an uncontrolled epidemic (the final epidemic size far far exceeds the herd immunity threshold, a phenomenon called epidemic overshoot), based on many epidemiological modeling studies. Again, this reemphasize the validity of the main conclusion of our work.

Line 163, change “having the world’s strictest COVID-19 policies (‘zero-COVID’)” to “the strict ‘zero-COVID’ policy”. As suggested previously, please use accurate and objective language.

Fixed.

Line 225, 202 should be 2020.

Fixed.

REVIEWERS' COMMENTS

Reviewer #3 (Remarks to the Author):

The authors have addressed my comments sufficiently. Below are some typos to be fixed.

Line 163, "84%" should be "82.4%".

Line 233, "intrinsic" should be "realized".

June 16, 2023

Reviewer comments are in *italics*, and author responses are in normal font.

Reviewer #3

The authors have addressed my comments sufficiently. Below are some typos to be fixed.

Line 163, “84%” should be “82.4%”.

Line 233, “intrinsic” should be “realized”.

Thank you! We fixed the two typos.